**Data Availability Statement:** All dataset files are available from the OSF (https://osf.io/5c49f/).

**Funding:** This research was supported in part by JSPS (Japan Society for the Promotion of Science, https://www.jsps.go.jp/) KAKENHI Grant Numbers JP18K13370 (KY), JP23H03703 (KT), and

# How many categories are there in crossmodal correspondences? A study based on exploratory factor analysis

Yuka Ohtake[1,2]*, Kanji Tanaka[3], Kentaro Yamamoto[4]*

**1** Graduate School of Human-Environment Studies, Kyushu University, Fukuoka, Japan, **2** Japan Society for the Promotion of Science, Tokyo, Japan, **3** Faculty of Arts and Science, Kyushu University, Fukuoka, Japan, **4** Faculty of Human-Environment Studies, Kyushu University, Fukuoka, Japan

* yuka.y.ohtake@gmail.com (YO); yamamoto.kntr@hes.kyushu-u.ac.jp (KY)

## Abstract

Humans naturally associate stimulus features of one sensory modality with those of other modalities, such as associating bright light with high-pitched tones. This phenomenon is called crossmodal correspondence and is found between various stimulus features, and has been suggested to be categorized into several types. However, it is not yet clear whether there are differences in the underlying mechanism between the different kinds of correspondences. This study used exploratory factor analysis to address this question. Through an online experiment platform, we asked Japanese adult participants (Experiment 1: $N = 178$, Experiment 2: $N = 160$) to rate the degree of correspondence between two auditory and five visual features. The results of two experiments revealed that two factors underlie the subjective judgments of the audiovisual crossmodal correspondences: One factor was composed of correspondences whose auditory and visual features can be expressed in common Japanese terms, such as the loudness–size and pitch–vertical position correspondences, and another factor was composed of correspondences whose features have no linguistic similarities, such as pitch–brightness and pitch–shape correspondences. These results confirm that there are at least two types of crossmodal correspondences that are likely to differ in terms of language mediation.

## Introduction

People receive a variety of sensory information simultaneously in the natural environment, such as hearing a person's voice while observing his/her mouth movements during a conversation. This kind of multisensory information is processed independently through specialized sensory systems and then integrated to form a coherent percept [1]. Specific features of a unimodal stimulus can be associated with different features of other sensory modalities, which are referred to as crossmodal correspondences (for a review, see Spence [2]). For example, bright and dark appearances are often associated with high- and low-pitched sounds, respectively. Crossmodal correspondences are a common phenomenon experienced by many people,

JP21J20359 (YO). The funders had no role in study design, data collection and analysis, decision to publish, or preparation of the manuscript.

**Competing interests:** The authors have declared that no competing interests exist.

unlike synesthesia, and have been observed between stimulus features across various sensory modalities, including touch, smell, and taste [3–5]. In the present study, we focused on cross-modal correspondences between auditory and visual stimulus features, which have been extensively studied previously.

Crossmodal correspondence has been demonstrated by using various tasks. Mudd [6] investigated the extent to which four sound dimensions (frequency, intensity, duration, and direction) were associated with horizontal/vertical spatial positions by asking participants to plug a peg into a position on a panel that they thought the sound stimulus would represent. The study revealed that higher frequencies and louder sounds tend to be plotted higher on the vertical axis, which was later interpreted as a correspondence between pitch/loudness and spatial position. Marks [7] examined the correspondence between sound and visual brightness by asking participants to manipulate the pitch/loudness of pure tones until they felt that the tone matched a gray square of different luminance. The participants matched the higher-pitched and louder tones to a brighter square. This finding is supported by a later study by Marks, Hammel, and Bornstein [8], who used a two-alternative forced-choice procedure and revealed that almost all participants over the age of four years match a high-pitched tone to bright light and low-pitched tone to dim light. In addition to these direct-matching tasks, the speeded classification paradigm has also been widely used in recent years to examine crossmodal correspondences (see [2]). In this paradigm, participants are asked to judge the feature of a target stimulus as quickly as possible while ignoring an irrelevant stimulus that is presented simultaneously. Reaction time is shorter when the feature of the irrelevant stimulus matches that of the target stimulus in terms of crossmodal correspondences compared with when they are inconsistent [7, 9–11]. These findings suggest that crossmodal correspondences, to some extent, operate automatically.

Crossmodal correspondences have also been studied in the context of child development and cross-linguistic comparison. Evidence has suggested that preverbal infants show the same crossmodal correspondences as adults [12–15]. Although these results imply that crossmodal correspondences may not be acquired through learning or experience, Haryu and Kajikawa [16] reported that 10-month-old children do not show pitch–size correspondences, although they can associate between pitch and brightness. Thus, not all kinds of correspondences are present in early development. It has been suggested that the strength of the space–pitch association may be susceptible to language use [17–19]. For example, adult Dutch speakers show a stronger height–pitch association than a thickness–pitch association, whereas adult Turkish speakers who have a thickness metaphor in language show the opposite tendency [17]. These findings indicate that language plays an important role in some kinds of crossmodal correspondences.

Spence [2] summarized a variety of crossmodal correspondences between visual and auditory features and their influence on information processing and categorized them into three principal types: structural, statistical, and semantic correspondences. Structural correspondences arise from neural connections between sensory systems or from common processing systems or mechanisms. The correspondences that occur between related features in the magnitude domain may be included in this type—they are represented by a generalized system in the brain [20]. Statistical correspondences are based on statistical regularities or co-occurrences between stimulus features in the environment [21]. For example, larger objects tend to produce lower-pitch sounds than smaller ones [22, 23], and such natural correlations may be learned to form the correspondence between pitch and size. Semantic correspondences are also acquired through learning but are primarily mediated by language (see also [24]). Typically, the features described with the same adjectives, such as "high" and "low," may be associated through their linguistic consistency.

These distinctions are important to understand the mechanisms of audio-visual crossmodal correspondences. Notably, there remain some limitations. First, the three types of crossmodal correspondences are not necessarily exclusive, and some correspondences may belong to more than one type. This makes it difficult to determine which type a given pair of auditory and visual features falls under. For example, the correspondence between pitch and elevation could be explained by either the internalization of natural statistics or the use of the same words, or both [2]. Second, the categorization is based on the difference in how each crossmodal correspondence can occur, with little consideration for the direct relations between them. Thus, the distinction may change if a new possible mechanism is found. Indeed, it has been suggested that some correspondencies between complex stimuli, such as music and color, are mediated by emotion [25, 26], referred to as the fourth mechanism for audiovisual correspondences [27]. These limitations may be due, at least in part, to the fact that each kind of crossmodal correspondence has been separately examined, resulting in ambiguity of their commonality or consistency.

We addressed the above queries by using exploratory factor analysis to examine the underlying structure of subjective ratings for different kinds of crossmodal correspondences. Factor analysis is a statistical method that is used to explore the underlying structure of a set of variables and to identify the latent factors that explain the patterns of correlation or variation among these variables. Although crossmodal correspondences are a common phenomenon that many people experience, previous studies have reported that there are differences across cultures and individuals in the patterns or strengths of correspondences [17, 19, 28, 29]. We used factor analysis to identify the underlying mechanisms of crossmodal correspondences based on such variation. The present study included some correspondences that were not listed in Spence (2011) or were not classified as a single type in his list. Therefore, we used exploratory factor analysis rather than confirmatory factor analysis to explore latent factors. Specifically, we focused on the correspondences between basic stimulus dimensions in the auditory and visual domains, which have been well demonstrated in the literature, to test whether there is a clear distinction between crossmodal correspondences, as suggested by Spence [2]. In Experiment 1, using a pair of visual stimuli and one auditory stimulus, we asked participants to rate which visual stimulus matched the auditory stimulus better. Experiment 2 presented a pair of auditory stimuli and one visual stimulus to the participants, who rated which auditory stimulus matched the visual stimulus better.

## Experiment 1

### Methods

**Participants.** We used the Yahoo! JAPAN Crowdsourcing website to recruit participants. Although there are no strict rules regarding the appropriate sample size in exploratory factor analysis, it is generally suggested that a participant-to-variable ratio of 5:1 to 10:1 is required [e.g., 30]. Thus, we recruited 200 participants, allowing for the possibility of excluding some samples and items during the analysis process. Participants were not screened at recruitment, but check questions were asked after the main task to ensure that they had performed the task correctly. In total, 178 participants (140 male; mean age = 49.0 ± 10.6 years) completed the online experiment. An additional 21 participants completed the experiment but were excluded from the analysis because they did not click the play buttons to listen to the auditory stimuli in the previewing phases ($N$ = 13), could not distinguish the auditory features of the stimuli in the check questions ($N$ = 5), or both ($N$ = 3). Participants received 30 Japanese yen worth of shopping points for completing the experiment.

**Ethics statement.** The experiment was conducted in April 2021, with special care taken to ensure the anonymity of the data. The authors did not have access to any information that

could identify individual participants during or after data collection, either at the recruitment site or at the online experimental platform. Informed consent was obtained carefully. In particular, the first screen of the experiment provided an overview of the experiment and the anonymity of the data and stated that no information that could identify individuals would be included in the data analysis and publication. Only participants who checked the box at the bottom of the screen indicating that they fully understood the experiment and agreed to participate in the study proceeded to the experiment. Our study was approved by the ethical committee of the Faculty of Human-Environment Studies, Kyushu University (No. 2020–027).

**Apparatus and stimuli.** We used the Gorilla Experiment Builder (www.gorilla.sc; [31]) to create and host the experiment. We asked the participants to use their own computers and web browsers to access the platform and perform the online experiment. Images and tones were presented as stimuli to participants during the experimental task. They were instructed to use their earphones or headphones to listen to the auditory stimuli. No instructions as to the observational environment were given to the participants. When the size of their browser window was smaller than the image resolution, the image was maximized within the window while maintaining the aspect ratio.

The auditory stimuli were two pairs of pure tones, one with different loudness levels (loud vs. soft) and the other with different pitch levels (high vs. low pitch). These stimuli were created by modulating the amplitude or frequency of a 625-Hz standard tone, which was used for volume control at the beginning of the experiment. Specifically, the amplitudes of the loud and soft stimuli were 1.5 and 0.5 times as great as that of the standard tone, respectively. The frequencies of the high-pitched and low-pitched stimuli were 750 Hz and 500 Hz, respectively. These frequencies were chosen for their ease of discrimination and low level of discomfort. The duration of each tone was 1,000 ms, including 20 ms linear ramps at on- and off-set. The participants played the auditory stimuli by clicking the play button presented on their screens.

The visual stimuli were five pairs of objects, each pair of which differed in terms of brightness (bright vs. dark), vertical position (high vs. low position), size (large vs. small), shape (rounded vs. angular), or spatial frequency (high SF vs. low SF). Each stimulus pair was placed side by side on a white background to form a single image (881 pix × 493 pix). The stimuli were circular in shape except for the rounded and angular stimuli. The diameters were 240 pix for the large stimulus, 120 pix for the small stimulus, and 180 pix for the other stimuli. The high and low SF stimuli consisted of sinusoidal gratings with spatial frequencies of 0.1 cycle/pix and 0.033 cycle/pix, respectively, and were oriented 45˚ to the left. The other stimuli were uniform in luminance: white (with black contour) for the bright stimulus, black for the dark stimulus, and gray for all other stimuli. Each stimulus pair was aligned horizontally (470 pix from center to center), except for the pair with different positions, in which the stimuli were placed diagonally with a vertical distance of 180 pix. Following the findings that crossmodal correspondences are based on the relative difference between the stimuli [32, 33], these values of the stimulus features were chosen to ensure the relative differences within the pairs for both the visual and auditory stimuli.

**Procedure.** Participants were directed from the crowdsourcing website to the Gorilla experimental platform. After receiving a detailed explanation of the experiment and providing informed consent, they were presented with a three-minute instructional video for performing the task. The standard tone was then presented, and the participants were instructed to adjust the volume of their computer to a comfortable level and to make no change to the volume during the experiment.

In the main task, the correspondences between the two auditory and five visual features were measured using a Likert-type rating scale. Each trial consisted of a previewing phase followed by a rating phase. In the previewing phase, the participants were instructed to listen to

two tones paired in terms of different loudness or pitch levels and to imagine the designated visual features (i.e., either the brightness, vertical position, size, shape, or pattern) associated with each of them. The tones could be replayed by clicking the play buttons placed side by side on the center of the screen. Once the imagination component was done, the participants clicked a continue button to move on to the rating phase. In the rating phase, the tone button placed on the left side was presented along with the paired visual stimuli and the rating scale. The participants were asked to rate which side of the visual stimulus was better matched to the presented tone by clicking on the seven-point scale ranging from "extremely left" to "extremely right." After the first rating, the participants rated the other tone of the pair again in the same manner, followed by the next trial. Fig 1 illustrates examples of the displays presented in the previewing and rating phases.

Each participant completed 10 trials, combining the two pairs of tones and five pairs of visual stimuli. The trial order was randomized for each participant. There were four possible combinations of the spatial arrangements of the tones (i.e., play buttons) and visual stimuli. For the tones, the loud and high-pitched tone buttons were placed on the left side, whereas the soft and low-pitched tones were placed on the right side, or vice versa. For the visual stimuli, the dark, high position, large, rounded, and high SF stimuli were placed on the left side, whereas the bright, low position, small, angular, and low SF stimuli were placed on the right side, or vice versa. The participants were randomly assigned to one of four spatial arrangements.

After the main task, the participants were asked two questions to confirm whether they could distinguish the auditory features of the tones. For each question, the tone pair with different loudness or pitch levels was presented side by side in the same spatial arrangement as the main task. The participants judged which tone was louder or higher in pitch by clicking one of the buttons placed under each tone. The experiment took about ten minutes to complete.

## Results and discussion

Fig 2 shows the results of the rating scales. The loud tone was judged more often to match the large, angular, high position, bright, and low SF stimuli, whereas the soft tone was judged

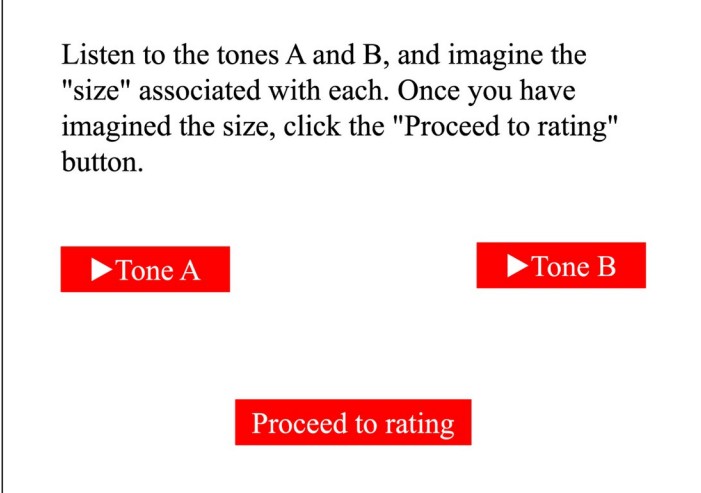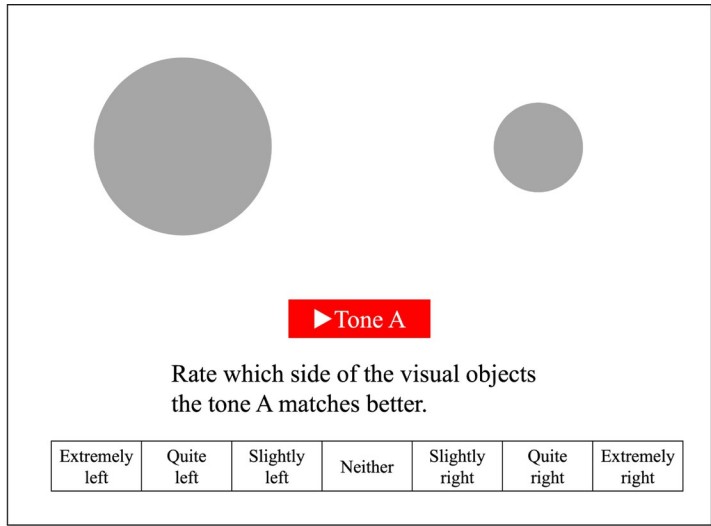

**Fig 1.** Examples of the displays presented in the previewing phase (left) and rating phase (right) in Experiment 1. The instructions and Likert scale were presented in Japanese.

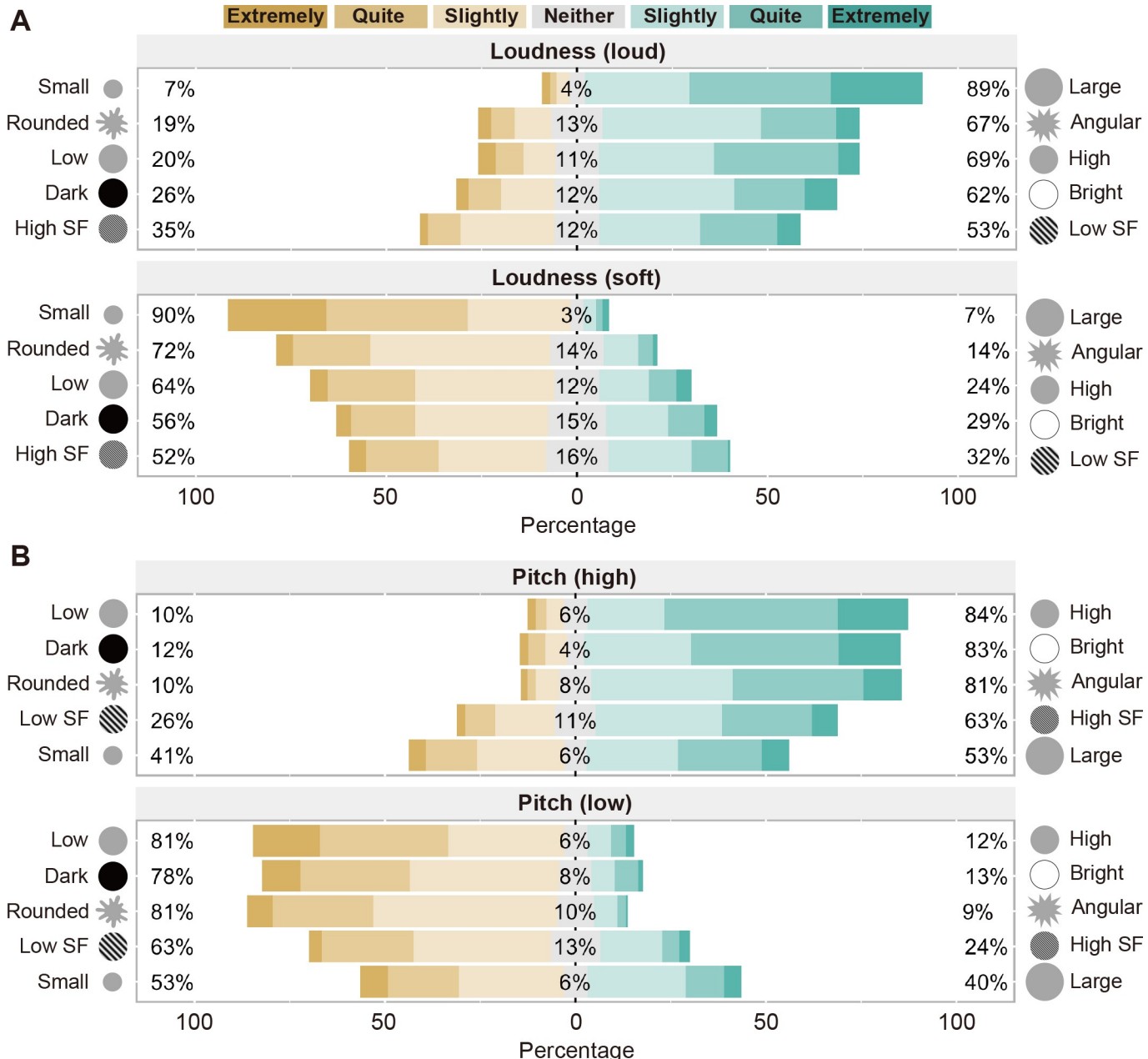

**Fig 2.** Results of the rating scales for crossmodal correspondences between loudness and visual features (A) and pitch and visual features (B) in Experiment 1. The numbers represent the percentages of participants who judged that either (or neither) of the paired visual stimuli matched the auditory feature at least slightly. This figure was created using the Likert package in R [34].

more often to match the small, rounded, low position, dark, and high SF stimuli. Moreover, the high-pitched tone was judged more often to match the high position, bright, angular, high SF, and large stimuli, whereas the low-pitched tone was judged more often to match the low position, dark, rounded, low SF, and small stimuli.

To examine the degree of correspondences, we designed the rating scales to be scored from -3 to 3, where positive and negative values indicated the identical and opposite directions of the correspondences described above, respectively. Table 1 shows the means and standard

**Table 1. Mean rating scores and results of one-sample *t*-tests for audiovisual correspondences.**

| Visual feature | Loudness | | | | Pitch | | | |
| --- | --- | --- | --- | --- | --- | --- | --- | --- |
| | Loud | | Soft | | High pitch | | Low pitch | |
| | *M* | *t* | *M* | *t* | *M* | *t* | *M* | *t* |
| Brightness | 0.57 | 4.96*** | 0.35 | 3.20* | 1.33 | 12.64*** | 1.05 | 10.55*** |
| Position | 0.76 | 6.63*** | 0.57 | 5.16*** | 1.50 | 14.87*** | 1.29 | 12.25*** |
| Size | 1.61 | 16.62*** | 1.67 | 17.85*** | 0.26 | 2.07 | 0.27 | 2.2 |
| Shape | 0.67 | 6.42*** | 0.81 | 9.16*** | 1.20 | 13.02*** | 1.09 | 13.46*** |
| Spatial frequency | 0.37 | 3.27* | 0.37 | 3.52* | 0.62 | 5.64*** | 0.61 | 5.92*** |

*adj $p < .05$

***adj $p < .001$

deviations of the rating scores and *t* values obtained from two-tailed one-sample *t*-tests. The *t*-tests with Bonferroni correction revealed that the rating scores were significantly higher than zero, except for the combination of pitch and size, suggesting the significant correspondences between various auditory and visual features. These results are consistent with those of previous studies that found crossmodal correspondences between various stimulus features using matching or speeded classification tasks. Meanwhile, the reported correspondence between pitch and size in previous studies [8, 9] was not significant. Rather, high (or low) pitch was matched better to large (or small) appearance, indicating an opposite trend from the previous reports.

To identify the potential factors, including their structure, underlying these correspondences, we conducted exploratory factor analysis using the rating scale data. Two factors were extracted based on the minimum average partial (MAP; [35]) criterion, and a scree plot supported the extraction. The results of the factor analysis using weighted least squares estimation and Oblimin rotation showed that the correspondence between loudness and spatial frequency (i.e., between loud tone and low SF and between soft tone and high SF) failed to reach loadings of 0.3 or higher, and the correspondence between pitch and spatial frequency (i.e., between high pitch and low SF and between low pitch and high SF) showed negative loadings, below -0.3. Thus, the same analysis was performed again without these items. The results are summarized in Table 2.

Factor 1 consisted of eight items related to the correspondences between loudness and size, pitch and vertical position, loudness and vertical position, and pitch and size. The Japanese language uses the words "large" and "small" to describe the difference in loudness as well as the difference in size, and "high" and "low" to describe the difference in pitch and vertical position. It also has linguistic similarities between loudness and vertical positions, as seen in expressions such as turning "up" and "down" the volume. Therefore, we considered that these correspondences were mainly mediated by language and named the factor "semantic correspondences," following the definition of Spence [2].

Factor 2 consisted of ten items related to the correspondences between pitch and brightness, loudness and brightness, pitch and shape, loudness and shape, and pitch and spatial frequency. These correspondences were unlikely to be mediated by language. Therefore, we named the factor "sensory correspondence." The internal consistencies assessed by the Cronbach's alpha coefficients were .80 and .78 for semantic and sensory correspondences, respectively, indicating acceptable reliabilities.

The results of the exploratory factor analysis indicated that two factors underlie the various kinds of correspondence between auditory and visual features. These factors were interpreted

**Table 2. Factor structures of crossmodal correspondences obtained in Experiment 1.**

| Factor 1: Semantic correspondence (α = .80) | Factor 1 | Factor 2 |
|---|---|---|
| Loudness–Size (Loud–Large) | **0.72** | 0.03 |
| Loudness–Size (Soft–Small) | **0.67** | 0.03 |
| Pitch–Position (High pitch–High position) | **0.62** | -0.11 |
| Loudness–Position (Loud–High position) | **0.61** | 0.09 |
| Pitch–Position (Low pitch–Low position) | **0.56** | -0.09 |
| Loudness–Position (Soft–Low position) | **0.56** | 0.12 |
| Pitch–Size (Low pitch–Small) | **0.46** | -0.09 |
| Pitch–Size (High pitch–Large) | **0.45** | -0.10 |
| Factor 2: Sensory correspondence (α = .78) | Factor 1 | Factor 2 |
| Pitch–Brightness (High pitch–Bright) | 0.09 | **0.73** |
| Pitch–Brightness (Low pitch–Dark) | -0.06 | **0.67** |
| Pitch–Shape (Low pitch–Round) | 0.00 | **0.59** |
| Pitch–Shape (High pitch–Angular) | 0.00 | **0.58** |
| Loudness–Brightness (Loud–Bright) | -0.04 | **0.58** |
| Loudness–Shape (Loud–Angular) | 0.03 | **0.50** |
| Loudness–Brightness (Soft–Dark) | -0.14 | **0.47** |
| Loudness–Shape (Soft–Round) | 0.09 | **0.35** |
| | | Correlation: .02 |

*Note.* Factor loadings above .30 are in bold.

as linguistically mediated and sensory-based correspondence. The correspondence between pitch and size was grouped into semantic correspondence, although they have no clear linguistic similarity between them in Japanese. The rating scores indicated the opposite trend from the previous study, suggesting the possibility that unexpected influences could have caused the semantic interaction between pitch and size during the experiment.

In this experiment, the degree of correspondence was judged based on auditory stimuli. That is, in the rating phase, the tones were presented individually to allow the participants to rate which of the visual stimuli was better matched to the tone. This led participants to make two ratings for each correspondence, such as matching a loud or soft tone with a pair of different visual sizes. However, these items of the same correspondence were grouped into the same factor and loaded on it to a similar extent (Table 2). This suggests that participants rate the features of the same correspondence on a similar basis. Meanwhile, the participants' judgments could have been based on auditory stimuli themselves, which might have affected the results. When the visual features to be matched were the same, the items were grouped into the same factor regardless of whether the targeted auditory feature was loudness or pitch, indicating that visual features had a greater impact on the emergence of the correspondences. To clarify the influence, we conducted Experiment 2, in which the degree of correspondences was judged based on visual stimuli.

## Experiment 2

### Methods

**Participants.** As in Experiment 1, we used the Yahoo! JAPAN Crowdsourcing website to recruit participants. A total of 160 participants (48 female, 111 male, 1 other; mean age = 47.7 ± 11.1 years) completed the online experiment. An additional 41 participants completed the experiment but were excluded from the analysis because they did not click the play

buttons to listen to the auditory stimuli in the rating phases (N = 29), could not distinguish the auditory features of the stimuli in the check questions (N = 3), or both (N = 9). We conducted the experiment in July 2021, taking special care to ensure the anonymity of the data and to obtain informed consent, as we did in Experiment 1.

**Apparatus, stimuli, and procedure.** The apparatus, stimuli, and procedure were the same as in Experiment 1, except for the following changes. In the previewing phase, one of the visual stimulus pairs was presented on the screen, and the participants were instructed to imagine the designated auditory features (i.e., loudness or pitch) associated with each of them. In the rating phase, the visual stimulus that had been placed on the left side was presented along with the tone pair and rating scale. Each tone was presented above each end of the scale. The participants were asked to rate which side of the tone was better matched to the presented visual stimulus by clicking on the seven-point scale ranging from "extremely left" to "extremely right." After the first rating, the visual stimulus was replaced by the other one of the pair, and the rating was performed again in the same manner. Fig 3 gives examples of the displays presented in the previewing and rating phases.

## Results and discussion

The results of the rating scales are shown in Fig 4. The large, angular, low SF, high position, and bright stimuli were judged more often to match the loud tone, whereas the small, rounded, high SF, low position, and dark stimuli were judged more often to match the soft tone. Moreover, the angular, bright, high position, small, and high SF stimuli were judged more often to match the high-pitched tone, whereas the dark, large, low position, rounded, and low SF stimuli were judged more often to match the low-pitched tone. These trends were the same as those in Experiment 1, except for the correspondence between pitch and size.

The rating scales were then scored from -3 to 3 to submit to two-tailed one-sample *t*-tests. The means and standard deviations of the rating scores and *t* values are shown in Table 3. The *t*-tests with Bonferroni correction revealed that the rating scores were significantly higher than zero for the combinations of all visual features and auditory pitch, whereas the rating scores were not significantly different from zero for the combinations of several visual features (i.e.,

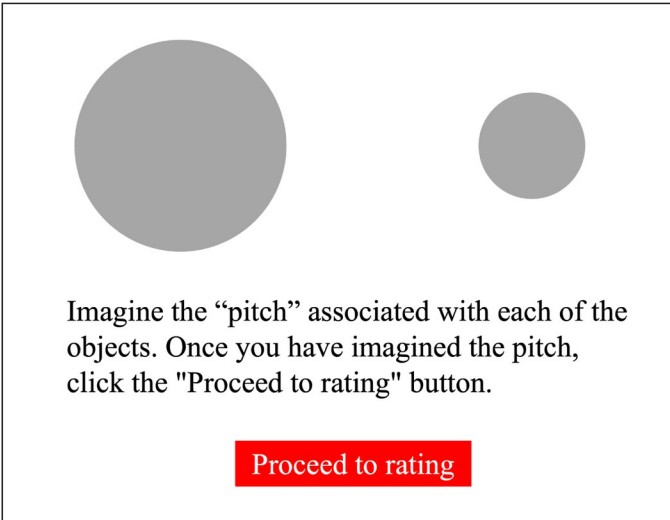
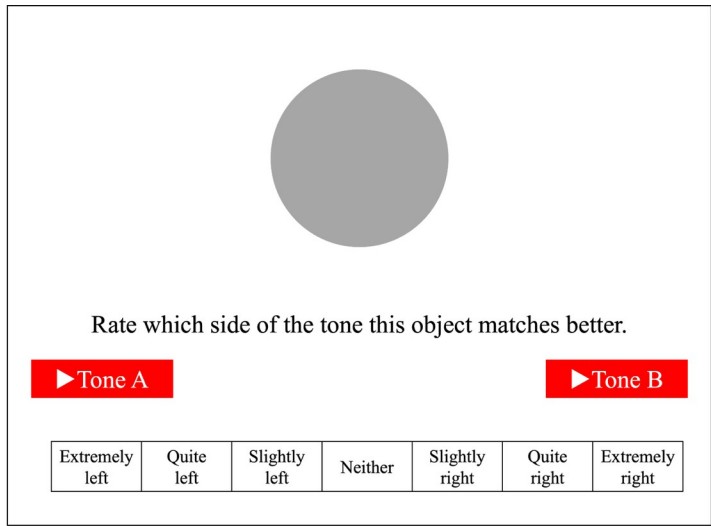

**Fig 3.** Examples of the displays presented in the previewing phase (left) and rating phase (right) in Experiment 2. The instruction and Likert scale were presented in Japanese.

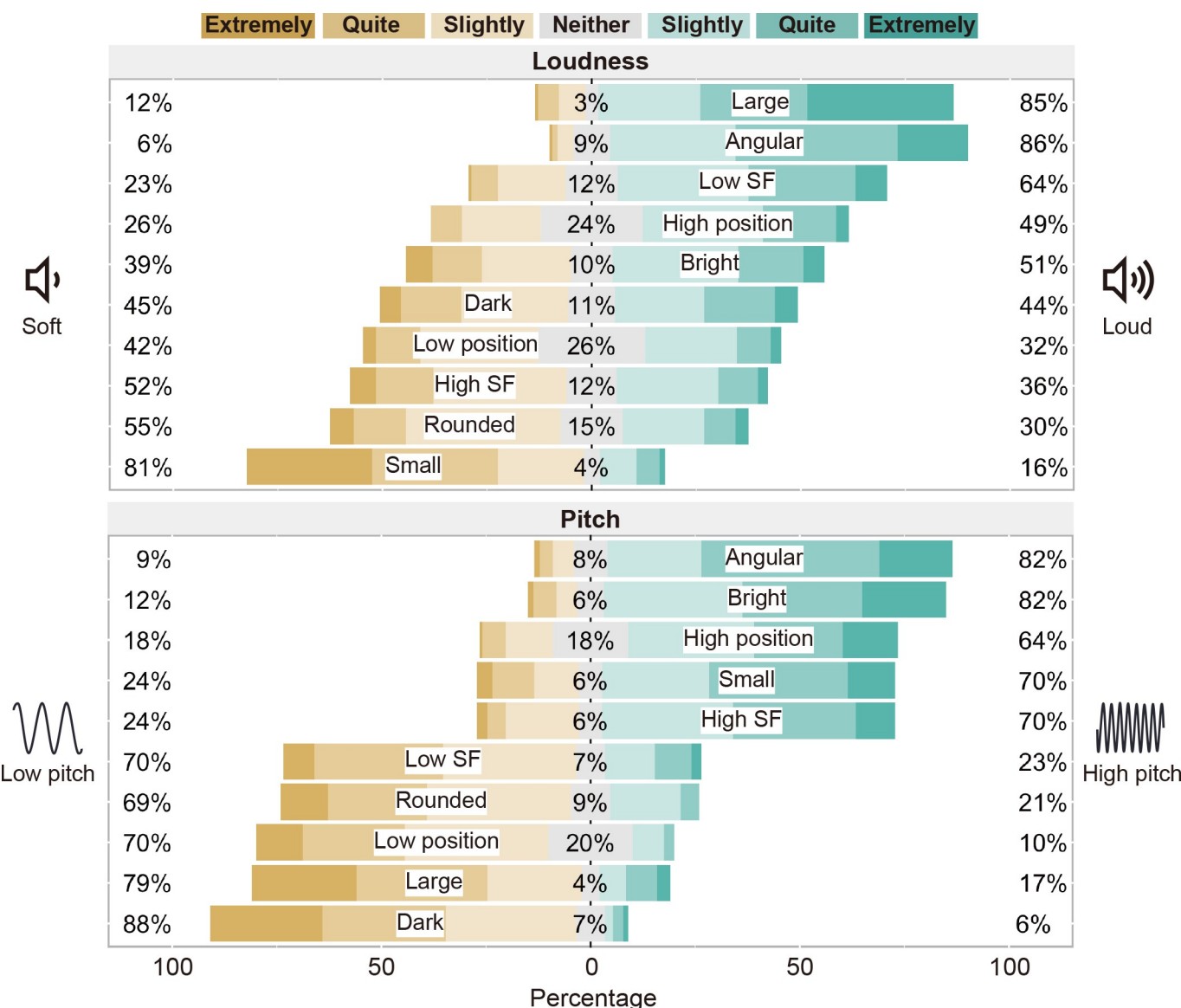

**Fig 4. Results of the rating scales for crossmodal correspondences between loudness and visual features and between pitch and visual features in Experiment 2.** The numbers represent the percentages of participants who judged that either (or neither) of the paired auditory stimuli matched the visual feature at least slightly.

bright, dark, low position, and high SF stimuli) and loudness. This finding was different from the results of Experiment 1, which showed significant correspondences between all visual features and loudness. Furthermore, unlike Experiment 1, the significant correspondence between size and pitch was shown in the same direction as in previous studies, where large (or small) appearance was matched better to low (or high) pitch. This difference means that judgments of crossmodal correspondences can be influenced by experimental procedures, such as the difference in the modality of the stimuli presented during the previewing phase.

We examined the relations among the correspondences using exploratory factor analysis. Two factors were extracted based on the MAP criterion, and a scree plot supported the extraction. The analysis using weighted linear squares estimation and Oblimin rotation revealed that

**Table 3. Mean rating scores and results of one-sample *t*-tests for audiovisual correspondences.**

| Visual feature | Loudness | | Pitch | |
|---|---|---|---|---|
| | *M* | *T* | *M* | *t* |
| Bright | 0.13 | 0.98 | 1.31 | 11.76*** |
| Dark | -0.03 | 0.19 | 1.60 | 16.06*** |
| High position | 0.39 | 3.92** | 0.88 | 7.88*** |
| Low position | 0.13 | 1.26 | 1.04 | 11.06*** |
| Large | 1.63 | 14.15*** | 1.29 | 9.94*** |
| Small | 1.47 | 12.02*** | 0.84 | 6.38*** |
| Rounded | 0.35 | 3.10* | 0.90 | 8.37*** |
| Angular | 1.50 | 16.83*** | 1.45 | 14.09*** |
| High SF | 0.28 | 2.34 | 0.84 | 7.18*** |
| Low SF | 0.74 | 6.75*** | 0.79 | 6.56*** |

*adj $p < .05$

**adj $p < .01$

***adj $p < .001$

the correspondence between loudness and shape (i.e., between loud tone and angular shape and between soft tone and round shape) and a part of the correspondence between loudness and vertical position (i.e., between soft tone and low position) failed to reach loadings of 0.3 or higher. Moreover, the correspondence between loudness and brightness (i.e., between loud tone and bright stimulus and between soft tone and dark stimulus) showed negative loadings below -0.3. Thus, we performed the analysis again with those items removed. The results are summarized in Table 4.

Factor 1 consisted of nine items, including the loudness–size and pitch–vertical position correspondences, whereas factor 2 consisted of eight items, including the pitch–brightness and pitch–shape correspondences. Thus, we named the factors semantic and sensory correspondences, respectively, as in Experiment 1. Their Cronbach's alpha coefficients were .73 and .73, respectively, indicating acceptable reliabilities.

Consistent with Experiment 1, the results of Experiment 2 indicated that the ten sets of audiovisual correspondences used in this experiment consistently accounted for the presence of semantic and sensory correspondence. Despite the similarities in the way the items were grouped, we found some differences between the experiments. Particularly, the correspondence between pitch and size, which was loaded on semantic correspondence in Experiment 1, was loaded on sensory correspondence in this experiment. The correspondence between pitch and size was judged in the opposite direction from Experiment 1, which suggests that participants associate stimulus features differently according to whether they judge it based on visual or auditory stimuli. This may also be why the correspondences between some visual features and loudness observed in Experiment 1 were not significant and were not grouped into either semantic or sensory correspondences in Experiment 2.

## General discussion

This study examined the different kinds of crossmodal correspondences in the audiovisual domain to find the distinctions among them. The results of the two online experiments revealed that two factors underlie the correspondence judgments between the two auditory and five visual features used. The correlation between the two factors was relatively low in both experiments, indicating that the two factors are relatively independent or distinct from

**Table 4. Factor structures of crossmodal correspondences obtained in Experiment 2.**

| Factor 1: Semantic correspondence (α = .73) | Factor 1 | Factor 2 |
|---|---|---|
| Loudness–Size (Loud–Large) | **0.83** | -0.07 |
| Loudness–Size (Soft–Small) | **0.80** | 0.01 |
| Pitch–Position (High pitch–High position) | **0.47** | 0.18 |
| Pitch–Position (Low pitch–Low position) | **0.42** | 0.20 |
| Loudness–Spatial frequency (Loud–Low SF) | **0.39** | 0.12 |
| Loudness–Spatial frequency (Soft–High SF) | **0.39** | -0.01 |
| Loudness–Position (Loud–High position) | **0.30** | -0.03 |
| Factor 2: Sensory correspondence (α = .73) | Factor 1 | Factor 2 |
| Pitch–Size (Low pitch–Large) | -0.09 | **0.62** |
| Pitch–Brightness (Low pitch–Dark) | 0.15 | **0.58** |
| Pitch–Size (High pitch–Small) | -0.06 | **0.57** |
| Pitch–Spatial frequency (Low pitch–Low SF) | -0.14 | **0.56** |
| Pitch–Brightness (High pitch–Bright) | 0.14 | **0.53** |
| Pitch–Spatial frequency (High pitch–High SF) | 0.01 | **0.48** |
| Pitch–Shape (High pitch–Angular) | 0.15 | **0.35** |
| Pitch–Shape (Low pitch–Round) | 0.13 | **0.30** |
| | | Correlation: .16 |

*Note.* Factor loadings above .30 are in bold.

each other. Although the components of each factor were slightly different between the experiments, the correspondences between features expressed in common Japanese linguistic terms, such as the loudness–size and pitch–vertical position correspondences, were consistently grouped into the same factor. These results suggest the existence of at least two types of crossmodal correspondences, which differ in terms of their semantic mediation. Moreover, some correspondences differed in terms of direction and classification type, depending on the procedure of correspondence judgments. Our results provide insights into the fundamental nature of crossmodal correspondences.

Our findings confirm that there are two kinds of crossmodal correspondences. Whereas Spence [2] had categorized crossmodal correspondence into the three principal types of structural, statistical, and semantic correspondences, our results showed a distinction only between semantic and other correspondences. This indicates the absence of a clear distinction between structural and statistical correspondences, at least in terms of the subjective evaluation of crossmodal correspondences. However, our results do not rule out the possibility that neural connections or learning of natural regularities underlie some crossmodal correspondences. Rather, these mechanisms may not be completely independent, even though they are triggered differently, as sensory experience may influence the development of neural structures [36, 37]. This may be the reason why the distinction between structural and statistical correspondences was not observed in the present study. Another possibility is the incorporation of what initially had been structural or statistical correspondences into semantic correspondence during language acquisition. Future studies are needed to determine how the distinctions of crossmodal correspondences change throughout language acquisition.

Our study also provides some implications for the mechanism of each kind of crossmodal correspondence. In both Experiments 1 and 2, we found that the loudness–size, pitch–vertical position, and loudness–vertical position correspondences were grouped into Factor 1, whereas the pitch–brightness and pitch–shape correspondences were grouped into Factor 2. Based on

the linguistic terms used to describe the stimuli, we considered the correspondences in Factor 1, but not those in Factor 2, to be mediated by language and named them as semantic and sensory correspondences, respectively. This is consistent with previous findings that pitch–brightness and pitch–shape correspondences can be observed even in preverbal infants [13, 16]. However, in Experiment 2, the correspondence between loudness and spatial frequency, which does not seem to be mediated by language, was categorized as semantic correspondence, whereas the correspondence between pitch and spatial frequency, whose stimuli can be described as "high" and "low," was categorized as sensory correspondence. These results may be because the words "thin" and "thick" are more commonly used in Japanese than the words "high" and "low" to describe stripe patterns that differ in spatial frequency [38]. Since thickness represents the distance between two sides, spatial frequency may have been semantically associated with loudness, for which the words "small" and "large" are used. Indeed, the high and low spatial frequency stimuli were judged more often to match the soft and loud tones, respectively, in both experiments. However, the high and low spatial frequency stimuli were judged more often to match the high- and low-pitched tones, respectively, as previously reported. Thus, the correspondence between pitch and spatial frequency may not be primarily mediated by language.

Meanwhile, the correspondence between pitch and size was grouped into different factors across the experiments; it was categorized as semantic correspondence in Experiment 1 and as sensory correspondence in Experiment 2. Thus, the mechanisms that drive crossmodal correspondences vary depending on how the correspondence is judged. However, this result may be explained by differences in how participants interpret the stimulus features. Krugliak and Noppeney [39] reported that, contrary to a previously reported pattern of the correspondence between pitch and size, low pitch is associated with small size and high pitch, with large size. They suggested that observers are likely to have interpreted the stimuli of different sizes as being at different distances and have associated the change in distance across trials with the change in pitch based on the Doppler effect. In our study, we found the opposite pattern of the pitch–size correspondence only in Experiment 1, where the tones of different pitch or loudness levels were presented in pairs in the previewing phase. Because the loudness threshold of a sound varies depending on its frequency [40], the participants might have interpreted the tones of different pitches as having different loudness levels by comparing them during the previewing stage. This would have been particularly likely when the participants were instructed to imagine the size of the visual objects associated with each of the tones since size is expressed as "ookisa" in Japanese and the same word is used to describe the loudness of a tone. As a result, both the higher-pitched tones as well as the louder tones may have been judged to match the larger stimuli in Experiment 1. This may also explain the grouping of the pitch–size correspondence into the same factor as the loudness–size correspondence in Experiment 1. It is necessary to pay more attention to the perceived loudness of auditory stimuli when examining the correspondence between pitch and visual features.

In the experiments of this study, the rating phase was preceded by the previewing phase, in which a pair of tones (a pair of visual stimuli in Experiment 2) was presented side by side. This phase was included because crossmodal correspondences are supposed to be based on the relative rather than absolute differences between auditory and visual features [32, 33], and thus the presentation of only one of a pair of tones during the rating phase is insufficient for participants to match the auditory feature with the visual feature. To facilitate the correspondence judgments, participants were asked to imagine the designated visual (or auditory) features associated with each of a pair of tones (or visual stimuli) presented during that phase. Although the left-right arrangement of the pair of tones was randomized across participants, the loud and high-pitched (or soft and low-pitched) tones were placed on the same side throughout the

experiment. This may have led participants to make an association between loudness and pitch, which may have influenced the judgment of crossmodal correspondences. Further research is needed to determine how the method of stimulus presentation affects the pattern of crossmodal correspondences.

The experimental method used in this study was novel in that participants rated several kinds of crossmodal correspondences within the same experiment. In previous studies, different kinds of crossmodal correspondence have been examined separately in different experiments, and thus, their relations have been difficult to ascertain. In contrast, by asking the same participants to make judgments about the correspondence between five visual and two auditory features, we were able to find new aspects of audiovisual crossmodal correspondence, such as differences in its strength and differences in its susceptibility to experimental procedures. This approach is consistent with Parise's [41] suggestion that it is important to examine individual differences in crossmodal correspondences by assessing whether different correspondences are similar within participants. Our method would be applicable for evaluating the (dis) similarity and relatedness between various kinds of crossmodal correspondences.

Nonetheless, the present study also has several limitations. The first relates to the generalizability of the results. Given that linguistic terms play a crucial role in crossmodal correspondences, the categorization of each correspondence may differ depending on the language used. Whether our results can be applied to participants from other language areas needs to be verified. Second, crossmodal correspondences were examined by asking participants to explicitly rate which of the paired stimuli was better matched to the stimulus in another modality. Thus, our results could have mainly reflected crossmodal correspondences that occurred at the decision level. Previous studies have suggested that some crossmodal correspondences occur at the perceptual level, based on the results of the speeded and unspeeded tasks [2, 10, 42–46]. Although the correspondences that occur at the perceptual level are likely to form the basis of explicit judgments, different factor structures might be observed when using such tasks. To clarify this, it would be needed to conduct controlled perceptual experiments on different kinds of crossmodal correspondences with the same participants. Different factor structures might be observed when using such tasks. Furthermore, it is also unclear whether the different types of crossmodal correspondences have different effects on performance in such behavioral tasks. Parise and Spence [42] found that the strength of the effect of crossmodal correspondences on the implicit association task was comparable across the five audiovisual correspondences they used. However, Barbosa Escobar, Velasco, Byrne, and Wang [47] showed that some correspondences between visual textures and temperatures observed in an explicit test were not found in the expected direction in the implicit association test, suggesting that explicit and implicit tasks may investigate different aspects of crossmodal correspondences. Future studies should also address these issues.

## Supporting information

**S1 Fig. Examples of the visual stimuli used in Experiment 1 and 2.**
(PDF)

## Author Contributions

**Conceptualization:** Yuka Ohtake, Kanji Tanaka, Kentaro Yamamoto.

**Data curation:** Kentaro Yamamoto.

**Formal analysis:** Kentaro Yamamoto.

**Funding acquisition:** Yuka Ohtake, Kanji Tanaka, Kentaro Yamamoto.

**Investigation:** Yuka Ohtake, Kentaro Yamamoto.

**Methodology:** Yuka Ohtake, Kentaro Yamamoto.

**Project administration:** Kentaro Yamamoto.

**Supervision:** Kanji Tanaka, Kentaro Yamamoto.

**Visualization:** Kentaro Yamamoto.

**Writing – original draft:** Yuka Ohtake, Kentaro Yamamoto.

**Writing – review & editing:** Yuka Ohtake, Kanji Tanaka, Kentaro Yamamoto.

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
