## [Decision Letter · Decision Letter 0]

12 Jul 2023

PONE-D-23-11568How many categories are there in crossmodal correspondences?

A study based on exploratory factor analysisPLOS ONE

Dear Dr. Ohtake,

Thank you for submitting your manuscript to PLOS ONE. After careful consideration, we feel that it has merit but does not fully meet PLOS ONE’s publication criteria as it currently stands. Therefore, we invite you to submit a revised version of the manuscript that addresses the points raised during the review process.

We look forward to receiving your revised manuscript.

Kind regards,

Kyoshiro Sasaki, Ph.D.

Academic Editor

PLOS ONE

Journal Requirements:

Additional Editor Comments:

Thank you for submitting your work to PLOS ONE. I apologize for the delay in our review process. The evaluations provided by our first two reviewers were somewhat contradictory, prompting me to include a third reviewer. Before I delve into their individual feedback, I must inform you that your manuscript will require major revisions before we can consider it for publication.

Reviewer 1 has highlighted several key issues that need to be addressed, such as a more thorough comparison between your study and previous studies conducted by Cesare Parise. Reviewer 1 found it difficult to evaluate your manuscript in its current form but believes that careful revisions could render it suitable for publication.

Reviewer 2 expressed significant concerns regarding your methodology and provided a stringent evaluation of your manuscript. While I concur with the reviewer's primary concerns, I believe your manuscript can be greatly improved through additions, corrections, and adjustments to the assertiveness of your claims based on Reviewer 2's feedback.

Reviewer 3 suggested that your manuscript could benefit from more detailed explanations justifying your research aim and the use of exploratory factor analysis. Reviewer 3 also found your Methods section to be insufficiently detailed.

On a minor note, it appears that the first and last names of the third author on the cover page may be reversed (Yamamoto Kentaro -> Kentaro Yamamoto).

I eagerly anticipate the resubmission of your manuscript, which I trust will be greatly improved by the reviewers' feedback.

Reviewers' comments:

Reviewer's Responses to Questions

**Comments to the Author**

1. Is the manuscript technically sound, and do the data support the conclusions?

Reviewer #1: Yes

Reviewer #2: No

Reviewer #3: Partly

2. Has the statistical analysis been performed appropriately and rigorously? 

Reviewer #1: Yes

Reviewer #2: Yes

Reviewer #3: No

3. Have the authors made all data underlying the findings in their manuscript fully available?

Reviewer #1: Yes

Reviewer #2: Yes

Reviewer #3: Yes

4. Is the manuscript presented in an intelligible fashion and written in standard English?

Reviewer #1: Yes

Reviewer #2: Yes

Reviewer #3: Yes

5. Review Comments to the Author

Reviewer #1: I enjoyed reading this paper and think it tackles a relevant and important issue.

That said, there are a number of issues that need attention prior to making final decision. Nevertheless, I would be hopeful that a publishable paper would emerge following careful revision.

There are a couple of theoretical issues that deserve greater attention/consideration and also the references need work.

The work of Cesare Parise would seem highly relevant at several points in text, but is currently not cited.

Parise, C. V. (2016). Crossmodal correspondences: Standing issues and experimental guidelines. Multisensory Research, 29, 7-28. – have a nice comparison of different correspondences and supposed agreement intuitively between observers.

Parise, C. V., Knorre, K., & Ernst, M. O. (2014). Natural auditory scene statistics shapes human spatial hearing. Proceedings of the National Academy of Sciences of the USA, 111, 6104-6108. – Intriguing paper showing the pitch-elevation correspondence only present in nature for certain pitches.

Parise, C., & Spence, C. (2009). ‘When birds of a feather flock together’: Synesthetic correspondences modulate audiovisual integration in non-synesthetes. PLoS ONE 4(5): e5664. – this paper may be first to convincingly show perceptual level effects for audiovisual correspondences, ie using an unspeeded task.

Parise, C. V., & Spence, C. (2012). Audiovisual crossmodal correspondences and sound symbolism: A study using the implicit association test. Experimental Brain Research, 220, 319-333. – this paper may be relevant in showing 5 different audiovisual correspondences all give rise to IAT of essentially same magnitude. Would you expect different categories would give rise to different magnitude of behavioural effects?

The header doesn’t make sense in English

Line 24 ‘qualitative distinctions’ meaning what precisely?

Two important questions: What should we say about role of perceptual similarity in driving crossmodal correspondences, which you mention at one point (see line 95) but never return to.

How do you know you are comparing apples with oranges? In other words, how do you know pitch difference is equivalent to size difference or height difference etc. ? If correspondences, especially those involving pitch are relative Spence, C. (2019). On the relative nature of (pitch-based) crossmodal correspondences. Multisensory Research, 32(3), 235-265. DOI:10.1163/22134808-20191407.

Spence 2011 briefly introduces affectively-mediated correspondences, and returns to this as 4th mechanism of eg audiovisual correspondences later: Spence, C., & Sathian, K. (2020). Audiovisual crossmodal correspondences: Behavioural consequences & neural underpinnings. In K. Sathian & V. S. Ramachandran (Eds.), Multisensory perception: From laboratory to clinic (pp. 239-258). San Diego, CA: Elsevier.

41. Stein & Meredith doesn’t seem a good reference for speech perception, since they never studied speech, and virtually never studied perception

116. My recollection is that Velasco and others may have started to investigate relative strength , or intuitiveness, of different correspondences

280 ungrammatical

340-341 – more details needed

362-363 – seems too general a claim, surely you can only make claim about small subset of correspondences you actually studied?

426-428 – how much weight should be put on this one study that seems to go against a large body of research showing pitch-size larger lower mapping?

References need lots of work, inconsistent capitalization of journal titles throughout

Missing page range eg 487

Missing vol. no e.g 475

Incorrect capitalization e.g. 525

Reviewer #2: This study investigated how many categories there are in crossmodal correspondences and how the different types of correspondences differ qualitatively. They conducted two online experiments in which participants were asked to rate the degree of correspondence between two auditory and five visual features. The results of exploratory factor analyses showed that there are at least two categories of crossmodal correspondences, one of which appears to be mediated by language and the other not.

The paper addresses an important issue in the field of crossmodal correspondence research, namely the categorization of crossmodal correspondences. Although many studies have suggested that there are different types of crossmodal correspondences, few studies have systematically compared crossmodal correspondences between several different features using exactly the same experimental task (cf. Anikin & Johansson, 2019, https://doi.org/10.3758/s13414-018-01639-7 ). In this respect, this present study provides valuable data. However, there are several major concerns, which I describe below.

Major concerns:

1) The experimental method used in this study, subjective ratings, may not be appropriate for investigating the purpose of this study. Subjective ratings generally reflect the cumulative output of multiple stages of processing, including low-level sensory and intermediate perceptual stages, as well as later stages involved in decision making. Such a measure may be insensitive to the differences in the categories of crossmodal correspondences. For example, low-level sensory or perceptual processing is thought to be involved in structural and statistical correspondences, whereas high-level cognitive processing is involved in semantic correspondences (Spence, 2011). It is difficult to detect the differences between these different types of crossmodal correspondences based on subjective rating data.

Authors may find Parise (2016, https://doi.org/10.1163/22134808-00002502 ) and Zeljko, Kritikos, & Grove (2019, https://doi.org/10.3758/s13414-019-01668-w) helpful in discussing experimental methods for conducting crossmodal correspondence research.

2) The combination of factor analysis with the current rating task may also be inappropriate for investigating the purpose of this study. Since crossmodal correspondences are a common phenomenon experienced by the general population (i.e., small individual differences), and the authors tested correspondences that have been well demonstrated in the literature, it should have been predicted that most participants would respond similarly to all stimulus pairs. Factor analysis, which extracts the common variance from variables to model a smaller number of latent factors, is not useful in this situation.

3) I agree with the authors in that Factor 1 (in both experiments) may involve language mediation. However, I am not sure whether this means that participants spontaneously experienced semantic (language-mediated) correspondences for the items which comprising Factor 1, or whether they strategically used the linguistic label commonality to associate the auditory and visual features in trials with these items to meet the task demands. If the latter possibility is true, the variances on these trials may reflect the individual differences in the degree to which such a strategy is used, rather than in sensitivity to spontaneous semantic crossmodal correspondences.

Minor comments:

4) Lines 96-109 “These distinctions are important to understand the mechanisms of audio-visual crossmodal correspondences. Notably, there remain some limitations. First, the three types of crossmodal correspondences are not necessarily exclusive, and some correspondences may belong to more than one type. This makes it difficult to determine which type a given pair of auditory and visual features fall under. ... Second, the categorization is based on the difference in how each crossmodal correspondence can occur, with little consideration for the direct relations between them. Thus, the distinction may change if a new possible mechanism is found. ... These limitations may be due, at least in part, to the fact that each kind of crossmodal correspondence has been separately examined, resulting in ambiguity of their commonality or consistency.”

It is not clear to me that these limitations can be addressed by using the subjective rating task and factor analysis. If some of the different types of correspondence are non-exclusive in nature (which I believe they are), it would be difficult to separate the types using factor analysis.

5) Lines 431-436 “In our study, we found the opposite pattern of the pitch-size correspondence only in Experiment 1, where the tones of different pitch or loudness levels were presented in pairs in the previewing phase. Because the loudness threshold of a sound varies depending on its frequency [32], the participants might have interpreted the tones of different pitches as those of different loudness levels by comparing them during the previewing stage.”

Perhaps the spatial arrangement of the tones in the previewing phase caused the opposite pattern of pitch-size correspondence in Experiment 1. Because the loud and high-pitched tone buttons were placed on the left side, whereas the soft and low-pitched tone buttons were placed on the right side, or vice versa, participants may have processed the four types of tones in two categories, such as “loud and high-pitched tone” vs. “soft and low-pitched tone”, which may have led to the grouping of the pitch-size correspondence into the same factor as the loudness-size correspondence in Experiment 1.

6) Typo?

Line 242 “a screen plot” -> “a scree plot”

Reviewer #3: Summary

This study attempted to investigate whether there are any qualitative distinctions among the different kinds of crossmodal correspondences by using exploratory factor analysis. The authors asked Japanese adult participants to rate the degree of correspondence between two auditory (i.e., loudness and pitch) and five visual features (size, shape, brightness, spatial frequency, and position). Two experiments were performed using the same stimuli, but presentation orders of visual and auditory features were exchanged. The authors showed that two factors (semantic and non-semantic factors) underlie the subjective judgments of the audiovisual crossmodal correspondences and claimed that at least two types of crossmodal correspondences are likely to differ in language mediation.

Reviews

Investigating the qualitative distinctions between different types of cross-modal correspondences would be significant in clarifying the correspondence mechanisms. This study attempts to explore the question by performing two online experiments. However, it requires some revision before submission for publication. The three main concerns identified in this study are as follows. 1) Authors should identify the advantages of investigating several cross-modal correspondences mixed in one experiment rather than experimenting with each cross-modal correspondence separately. 2) The theoretical motivation for introducing the exploratory factor analysis should be written. 3) Details of stimuli presentation, experimental procedures, and instruction in online experiments should be explicitly written.

The following are the observations in detail.

Major points

1. Regarding the limitations of Spence’s categorization of crossmodal correspondence, the authors claim that “each kind of crossmodal correspondence has been separately examined, resulting in the ambiguity of their commonality or consistency.” Please clarify how the separate examination could lead to the ambiguity of their commonality or consistency and how the simultaneous examination, where the different types of crossmodal correspondences are presented, would eliminate the ambiguity of the commonality.

2. Please clarify the theoretical motivation for using exploratory factor analysis in this study, i.e., how exploratory factor analysis would contribute to understanding the qualitative distinctions between different types of cross-modal correspondence. As the authors state in the Introduction (lines 112-113), Spence has already classified various cross-modal correspondences into three types. The authors, therefore, already have a hypothesis regarding the distinction between correspondences. Exploratory factor analysis is intended to generate hypotheses, and as long as a clear hypothesis already exists, exploratory factor analysis is not considered appropriate. Please provide a convincing explanation for the use of exploratory factor analysis.

3. The stimulus description, experimental procedure, and experimental environment should be described so that the reliability and validity of the observed cross-modal responses can be understood. Details are given in the following fine points.

Minor points

1) Method: Line 125-126, please clarify the rationale for the number of participants and how the number of participants was decided. Is it correct to say that 199 participants participated in the experiment, and data of 21 participants were excluded from the analysis?

2) Please make explicit whether the authors check visual acuity and hearing for each participant in recruiting the participants.

3) Line 127, Please clarify how the authors knew that the 21 participants did not listen to the auditory stimuli.

4) Please clarify whether there was any reward for participating in the experiment.

5) Line 142, is there any instruction for the monitor size, observation distance, or lighting environment, or are they entirely up to the participants?

6) Line 142, the loudness of stimuli with different pitches should be equalized by adjusting the stimuli intensity because the loudness threshold of a sound varies depending on its frequency. Please clarify how the authors adjust the sound intensity for each pitch.

7) In lines 179 – 181, the authors asked participants to imagine the designated visual features (i.e., brightness, vertical position, size, shape, or pattern) associated with each. Please clarify the purpose of the procedure.

8) It would be beneficial if authors could present the stimuli example of pairs of rounded–angular shapes, high SF and Low SF objects, and high and low positioned circles in Figure 1.

9) In lines 206-210, the authors asked the participants to confirm whether they could distinguish the tones' auditory features. Please indicate the results of this experiment in belief.

10) Please indicate how much time it took from the start to the end of the experiment.

11) Results: In line 241, the authors wrote, "Two factors were extracted based on minimum average partial (MAP) criterion and a screen plot." Did the MAP and the screen plot extract the same number of factors? Where the two results differ, please state which result is preferred.

12) Discussion: In lines 387 - 390, please make a rational explanation of why the distinction between structural and statistical correspondences has yet to be extracted in present studies.

13) In lines 411–413, the authors wrote, "…that the words "thin" and "thick" are more commonly used by people in Japan than the words "high" and "low" to describe stripe patterns that differ in spatial frequency." Please give references that support the statement.

14) Lines 424–426, Please clarify how the participants’ interpretation has differed between the presentation order (auditory or visual stimuli first).

6. PLOS authors have the option to publish the peer review history of their article (what does this mean?). If published, this will include your full peer review and any attached files.

Reviewer #1: **Yes: **Charles Spence

Reviewer #2: No

Reviewer #3: No

---

## [Author Response · Author response to Decision Letter 0]

21 Sep 2023

Response to Reviewer #1

We greatly appreciate the time and effort that Reviewer #1 has put into reviewing our manuscript. Here, we summarize how we have addressed your suggestions and revised the manuscript. 

Comment #1: 

The work of Cesare Parise would seem highly relevant at several points in text, but is currently not cited.

Parise, C. V. (2016). Crossmodal correspondences: Standing issues and experimental guidelines. Multisensory Research, 29, 7-28.

 – have a nice comparison of different correspondences and supposed agreement intuitively between observers.

Parise, C. V., Knorre, K., & Ernst, M. O. (2014). Natural auditory scene statistics shapes human spatial hearing. Proceedings of the National Academy of Sciences of the USA, 111, 6104-6108. 

– Intriguing paper showing the pitch-elevation correspondence only present in nature for certain pitches.

Parise, C., & Spence, C. (2009). ‘When birds of a feather flock together’: Synesthetic correspondences modulate audiovisual integration in non-synesthetes. PLoS ONE 4(5): e5664. 

– this paper may be first to convincingly show perceptual level effects for audiovisual correspondences, ie using an unspeeded task.

Parise, C. V., & Spence, C. (2012). Audiovisual crossmodal correspondences and sound symbolism: A study using the implicit association test. Experimental Brain Research, 220, 319-333. 

– this paper may be relevant in showing 5 different audiovisual correspondences all give rise to IAT of essentially same magnitude. Would you expect different categories would give rise to different magnitude of behavioural effects?

Response: 

We sincerely appreciate your sharing of the highly relevant papers by Dr. Cesare Parise. Following your suggestion, we have included the papers in the revised manuscript as follows, where appropriate.

•　Parise (2016) suggests that there are a few studies that have examined individual differences in crossmodal correspondences, and it is important to assess whether different correspondences are similar within participants. Because this suggestion is in line with the aim of our study, we have added an explanation in the General Discussion (L501∼504). 

•　Parise, Knorre, and Ernst (2014) suggested that natural statistical mapping may play a crucial role in crossmodal correspondences. We have cited it in the Introduction of the revised manuscript (L91). 

•　Parise and Spence’s (2009) study is a good example of a perceptual level effect in unspeeded tasks, so we have cited it along with other relevant papers in the General Discussion (L513∼515).

•　In the present study, participants were asked to make explicit matching judgments; therefore, it is still unclear whether different types of crossmodal correspondences have different effects on performance in behavioral tasks such as the implicit association test used in Parise and Spence (2012). Indeed, Barbosa Escobar et al. (2023) showed that some correspondences observed in an explicit test were not found in the expected direction in the implicit association test, suggesting that explicit and implicit tasks may investigate different aspects of crossmodal correspondences. We have extended the discussion on this point (L520∼528).

Comment #2: 

The header doesn’t make sense in English

Response: 

Thank you for pointing this out. We have revised the header as “Exploring factors of crossmodal correspondences.”

Comment #3: 

Line 24 ‘qualitative distinctions’ meaning what precisely?

Response: 

We meant to say that it is not yet clear whether there are differences in the underlying mechanisms between the different kinds of correspondences. To make this clear, we have revised the description (L24). 

Comment #4: 

What should we say about role of perceptual similarity in driving crossmodal correspondences, which you mention at one point (see line 95) but never return to.

Response: 

We used this phrase to emphasize that the same labels are used across different modalities in different languages. Thus, we did not mean to imply that there is a specific perceptual similarity between auditory and visual modalities and that might influence crossmodal correspondences. Since the expression was misleading, we deleted the phrase in the revised manuscript (L96).

Comment #5: 

How do you know you are comparing apples with oranges? In other words, how do you know pitch difference is equivalent to size difference or height difference etc. ? If correspondences, especially those involving pitch are relative Spence, C. (2019). On the relative nature of (pitch-based) crossmodal correspondences. Multisensory Research, 32(3), 235-265. DOI:10.1163/22134808-20191407.

Response:

Thank you for sharing the critical study. As the study suggests, we assumed that crossmodal correspondences were based on the relative differences between the auditory and visual features. Thus, we created the stimuli with a focus on the relative differences, without precisely considering the absolute value or the magnitude of the difference for each feature. We cited Spence (2019) and added an explanation in the Apparatus and stimuli section (L191∼194).

Comment #6: 

Spence 2011 briefly introduces affectively-mediated correspondences, and returns to this as 4th mechanism of eg audiovisual correspondences later: Spence, C., & Sathian, K. (2020). Audiovisual crossmodal correspondences: Behavioural consequences & neural underpinnings. In K. Sathian & V. S. Ramachandran (Eds.), Multisensory perception: From laboratory to clinic (pp. 239-258). San Diego, CA: Elsevier.

Response:　

Thank you for mentioning the relevant paper. We have cited Spence and Sathian (2020) and added the explanation of the 4th mechanism (L108∼109).

Comment #7:

41. Stein & Meredith doesn’t seem a good reference for speech perception, since they never studied speech, and virtually never studied perception

Response: 

Thank you for pointing this out. Instead of Stein and Meredith (1993), we have cited Calvert, Spence, and Stein (2004) as a more appropriate reference (L41).

Comment #8:

116. My recollection is that Velasco and others may have started to investigate relative strength, or intuitiveness, of different correspondences

Response:　 

We found that Barbosa Escobar, Velasco et al. (2023) investigated different correspondences between visual textures and temperatures using the explicit task and the implicit association test and showed different results for some correspondences. As this finding is essential for understanding how different types of crossmodal correspondences influence performance in different tasks, we have cited this paper in the relevant text in the Discussion (L520∼528).

Comment #9:

280 ungrammatical

Response: 

Thank you for pointing this out. We have corrected the sentence (L304∼307). 

Comment #10:

340-341 – more details needed

Response: 

We considered that the difference in the modality of the stimuli presented during the previewing phase might have influenced the difference in results between the experiments. As the details are discussed in the General Discussion section, we have revised the description (L369∼370).

Comment #11:

362-363 – seems too general a claim, surely you can only make claim about small subset of correspondences you actually studied?

Response: 

As you pointed out, we admit that our claim in the submitted manuscript was somewhat generalized. We have revised the sentence (L390∼392). 

Comment #12:

426-428 – how much weight should be put on this one study that seems to go against a large body of research showing pitch-size larger lower mapping?

Response:　

Krugliak and Noppeney discussed their experimental results as being due to participants’ interpretation of their stimuli, so we did not think that their results contradicted the previous research showing pitch-size larger lower mapping. Since the opposite pattern of the pitch-size correspondence, observed in Experiment 1 of the present study, can also be interpreted as being caused by participants’ interpretation of the stimuli, we presented their study as an example of how the way stimuli are presented can influence crossmodal correspondences. To make this clear, we modified the descriptions in the General Discussion (L470∼474).

Comment #13:

References need lots of work, inconsistent capitalization of journal titles throughout 

Missing page range eg 487

Missing vol. no e.g 475

Incorrect capitalization e.g. 525

Response:

Thank you for your attention to detail. We have checked the format of the Reference and corrected the mistakes.

Response to Reviewer #2

We greatly appreciate the time and effort that Reviewer #2 took to review our manuscript. Here, we summarize how we have addressed your suggestions and revised the manuscript.

Comment #1: 

The experimental method used in this study, subjective ratings, may not be appropriate for investigating the purpose of this study. Subjective ratings generally reflect the cumulative output of multiple stages of processing, including low-level sensory and intermediate perceptual stages, as well as later stages involved in decision making. Such a measure may be insensitive to the differences in the categories of crossmodal correspondences. For example, low-level sensory or perceptual processing is thought to be involved in structural and statistical correspondences, whereas high-level cognitive processing is involved in semantic correspondences (Spence, 2011). It is difficult to detect the differences between these different types of crossmodal correspondences based on subjective rating data.

Authors may find Parise (2016, https://doi.org/10.1163/22134808-00002502 ) and Zeljko, Kritikos, & Grove (2019, https://doi.org/10.3758/s13414-019-01668-w) helpful in discussing experimental methods for conducting crossmodal correspondence research.

Response:　

Thank you for your suggestion and for sharing the important papers. We understand the importance of examining the low-level sensory or perceptual processing of crossmodal correspondences, and the perceptual or behavioral tasks have an advantage in distinguishing between higher (i.e., decisional) and lower (i.e., perceptual) levels of processing. We have outlined the limitations of the present study in the General Discussion of the revised manuscript (L510∼). However, a disadvantage of the behavioral tasks is that they take a long time to complete due to numerous repetitions, which makes it difficult for researchers to conduct controlled perceptual experiments on different kinds of crossmodal correspondences with the same participants. To focus on how crossmodal correspondences are shared within participants, we used the subjective rating task, which can be completed quickly compared to behavioral tasks, to investigate more combinations of auditory and visual stimuli. As a result, we were able to identify general trends, such as difference in intensity, and individual differences between participants. Furthermore, the correspondences that occur at the perceptual level are likely to form the basis of explicit judgments at the decision level. Therefore, we believe that the method used in this study was one of the appropriate methods to investigate the purpose of this study. To make this clear, we have cited the references you suggested and revised the discussion (L501∼ 504, L513∼519). 

Comment #2: 

The combination of factor analysis with the current rating task may also be inappropriate for investigating the purpose of this study. Since crossmodal correspondences are a common phenomenon experienced by the general population (i.e., small individual differences), and the authors tested correspondences that have been well demonstrated in the literature, it should have been predicted that most participants would respond similarly to all stimulus pairs. Factor analysis, which extracts the common variance from variables to model a smaller number of latent factors, is not useful in this situation.

Response:　

Thank you for raising this important question. As you pointed out, crossmodal correspondences are a common phenomenon that experienced by many people, but this does not mean that there is little individual variation. In fact, previous studies have reported that there are differences across cultures and individuals in the patterns or strengths of crossmodal correspondences (Doschield et al. 2020; Fernandez-Prieto et al., 2017; Occelli et al., 2013; Rusconi et al., 2006). In addition, the results of our experiments consistently extracted two factors, that could be reasonably interpreted as indicating that participants did not respond similarly to all stimulus pairs. We have added an explanation of this in the Introduction (L114∼120).

Comment #3:

I agree with the authors in that Factor 1 (in both experiments) may involve language mediation. However, I am not sure whether this means that participants spontaneously experienced semantic (language-mediated) correspondences for the items which comprising Factor 1, or whether they strategically used the linguistic label commonality to associate the auditory and visual features in trials with these items to meet the task demands. If the latter possibility is true, the variances on these trials may reflect the individual differences in the degree to which such a strategy is used, rather than in sensitivity to spontaneous semantic crossmodal correspondences.

Response:　 

We cannot rule out the possibility that the task demands may have influenced the correspondences in Factor 1. Due to the nature of semantic correspondences, it is difficult to completely disentangle this influence from other effects, even when behavioural tasks are used. However, it is worth noting that this study did not use linguistic questions such as "Which visual stimulus matches the high-pitched tone?", but instead used a method of direct matching of visual and auditory stimuli. Therefore, the impact of task demands should have been minimized as much as possible within the method we used.

Comment #4: 

Lines 96-109 “These distinctions are important to understand the mechanisms of audio-visual crossmodal correspondences. Notably, there remain some limitations. First, the three types of crossmodal correspondences are not necessarily exclusive, and some correspondences may belong to more than one type. This makes it difficult to determine which type a given pair of auditory and visual features fall under. ... Second, the categorization is based on the difference in how each crossmodal correspondence can occur, with little consideration for the direct relations between them. Thus, the distinction may change if a new possible mechanism is found. ... These limitations may be due, at least in part, to the fact that each kind of crossmodal correspondence has been separately examined, resulting in ambiguity of their commonality or consistency.”

It is not clear to me that these limitations can be addressed by using the subjective rating task and factor analysis. If some of the different types of correspondence are non-exclusive in nature (which I believe they are), it would be difficult to separate the types using factor analysis.

Response: 

We did not mean to suggest that the different types of crossmodal correspondences are inseparably similar. As Spence (2011) suggested, the three types of crossmodal correspondences may have different mechanisms such as neural connections, statistical regularities, or language. However, some crossmodal correspondences could have more than one of these mechanisms, which is why we said that they are not necessarily exclusive. In this case, participants’ responses can vary depending on which type of mechanism is more dominant. Factor analysis allows us to identify underlying common factors based on the differences in response tendencies. Therefore, we considered the factor analysis appropriate for classifying types of cross-modal correspondence. Furthermore, we found that the interfactor correlations in both experiments were relatively low; r = .02 in Experiment 1 and r = .16 in Experiment 2. This indicates that the two extracted factors are relatively independent or distinct from each other. We have added the explanations of the factor analysis and the independence of the two factors extracted in this study in the Introduction (L114∼116) and General Discussion (L407∼409).

Comment #5:

Lines 431-436 “In our study, we found the opposite pattern of the pitch-size correspondence only in Experiment 1, where the tones of different pitch or loudness levels were presented in pairs in the previewing phase. Because the loudness threshold of a sound varies depending on its frequency [32], the participants might have interpreted the tones of different pitches as those of different loudness levels by comparing them during the previewing stage.”

Perhaps the spatial arrangement of the tones in the previewing phase caused the opposite pattern of pitch-size correspondence in Experiment 1. Because the loud and high-pitched tone buttons were placed on the left side, whereas the soft and low-pitched tone buttons were placed on the right side, or vice versa, participants may have processed the four types of tones in two categories, such as “loud and high-pitched tone” vs. “soft and low-pitched tone”, which may have led to the grouping of the pitch-size correspondence into the same factor as the loudness-size correspondence in Experiment 1.

Response: 

Thank you for drawing our attention to the potential impact of the button layout, which we had not fully considered. However, we believe that its effect was either minimal or even negligible for the following reasons. First, in the rating phase of Experiment 2, two tone buttons were placed on the left and right as in the training phase of Experiment 1. If the horizontal placement of the tone buttons had influenced the results, we would expect the same effect in Experiment 2, which was not the case. Second, in both the experiments, the pitch-spatial frequency and loudness-frequency correspondences showed opposite patterns; the high-pitched and loud tones were judged to be associated with the high and low spatial frequencies, respectively. This result cannot be explained if the participants were making judgments in two categories: "loud and high-pitched tone" vs. "soft and low-pitched tone." However, the spatial arrangement may have influenced the results in a different way, we have added an explanation about this in the General Discussion section (L479∼491). 

Comment #6:

Typo? Line 242 “a screen plot” -> “a scree plot”

Response: 

Thanks for pointing out the typo. We have corrected it in the revised manuscript (L268).

Response to Reviewer #3

We greatly appreciate the time and effort Reviewer #3 dedicated to evaluating our manuscript. Here, we summarize how we addressed your suggestions and revised the manuscript.

Comment #1: 

Regarding the limitations of Spence’s categorization of crossmodal correspondence, the authors claim that “each kind of crossmodal correspondence has been separately examined, resulting in the ambiguity of their commonality or consistency.” Please clarify how the separate examination could lead to the ambiguity of their commonality or consistency and how the simultaneous examination, where the different types of crossmodal correspondences are presented, would eliminate the ambiguity of the commonality.

Response: 

Thank you for this important question. Previous studies have examined the relationships between different kinds of crossmodal correspondences by comparing the strength or statistical significance of the effects. However, since each kind of crossmodal correspondence has been separately examined in different experiments or tasks, individual differences or task differences may have influenced the difference in strength or statistical significance of them. Indeed, previous studies have shown that there are differences in patterns or strengths of crossmodal correspondences across cultures or individuals (Doschield et al. 2020; Fernandez-Prieto et al., 2017; Occelli et al., 2013; Rusconi et al., 2006), and across tasks (Barbosa Escobar et al., 2023). Therefore, we asked the same participants to make judgments about the various kinds of crossmodal correspondences. In addition, we used factor analysis to explore the commonality and underlying structure of the judgments and to identify the latent factors that explain the patterns of correlation or variation among them. This approach has not been done before, and with it we were able to find the new aspects of audiovisual crossmodal correspondences. We have added some explanations of this in the revised manuscript (L114∼120, L501∼504, L520∼528).　 

Comment #2:

Please clarify the theoretical motivation for using exploratory factor analysis in this study, i.e., how exploratory factor analysis would contribute to understanding the qualitative distinctions between different types of cross-modal correspondence. As the authors state in the Introduction (lines 112-113), Spence has already classified various cross-modal correspondences into three types. The authors, therefore, already have a hypothesis regarding the distinction between correspondences. Exploratory factor analysis is intended to generate hypotheses, and as long as a clear hypothesis already exists, exploratory factor analysis is not considered appropriate. Please provide a convincing explanation for the use of exploratory factor analysis.

Response: 

As we described in the response to your comment #1, we used factor analysis to explore the commonality and underlying structure of the judgments of crossmodal correspondences and to identify the latent factors that explain the patterns of correlation or variation among them. Because this study included some correspondences that were not listed in Spence (2011) or were not classified as a single type in his list, we used exploratory factor analysis rather than confirmatory factor analysis, which cannot be used without a clear classification hypothesis. To make this clear, we have added an explanation in the Introduction (L120∼123).

Comment #3:

Method: Line 125-126, please clarify the rationale for the number of participants and how the number of participants was decided. Is it correct to say that 199 participants participated in the experiment, and data of 21 participants were excluded from the analysis?

Response: 

Yes, it is correct. The 199 participants joined our experiment, and the data of 21 participants were excluded from data analysis. 

Although there are no strict rules regarding the appropriate sample size in exploratory factor analysis, it is generally suggested that a participant-to-variable ratio of 5:1 to 10:1 is required (e.g., Hair, Black, Babin, & Anderson, 2010). Therefore, we recruited 200 participants for this study, allowing for the possibility of excluding some samples and items during the analysis process. We have added this information in the method (L136∼139). 

Comment #4:

Please make explicit whether the authors check visual acuity and hearing for each participant in recruiting the participants.

Response:

We did not check the visual acuity of the participants because it was supposed that they would be able to discriminate the visual stimuli if they had sufficient visual acuity to read the text in the instructions. Although hearing acuity was not checked beforehand, we presented two check questions at the end of the experiment to confirm whether participants could distinguish the auditory features of the tones (i.e., loudness or pitch levels), and only those who answered both questions correctly were included in the analysis. Since it is difficult to answer this question correctly without normal vision and hearing, we believed that participants who answered the check questions accurately had no problems with their vision or hearing. We have added the explanation about this in the method (L139∼141).

Comment #5:

Line 127, Please clarify how the authors knew that the 21 participants did not listen to the auditory stimuli.

Response:

Whether or not the participant listened to the auditory stimuli was verified by whether the play buttons were clicked or not. We have revised the description to clarify this point. (L143∼144, L321∼322).

Comment #6:

Please clarify whether there was any reward for participating in the experiment.

Response:

The participants received rewards through shopping points on the crowdsourcing site, equivalent to 30 Japanese yen. We have added this information in the procedure section (L146∼147).

Comment #7:

Line 142, is there any instruction for the monitor size, observation distance, or lighting environment, or are they entirely up to the participants?

Response:

As we described in the method, participants were instructed to use their own computers and web browsers to access the experimental platform, and to use their earphones or headphones to listen to the auditory stimuli. No instructions as to the observation environment were given to the participants. We did so because it has been suggested that crossmodal correspondences are based on the relative difference between the stimuli (Brunetti et al., 2018; Spence, 2019), and the absolute size of the stimulus was supposed to have little or no effect on the results. We have added the explanation about this in the Method (L165∼166, L191∼194).

Comment #8:

Line 142, the loudness of stimuli with different pitches should be equalized by adjusting the stimuli intensity because the loudness threshold of a sound varies depending on its frequency. Please clarify how the authors adjust the sound intensity for each pitch.

Response: 

We agree that it is important to match the perceived loudness between stimuli. However, the online platform used for our experiment did not allow for precise manipulation of sound volume or rigorous measurement of it. In addition, preliminary observations did not reveal much difference in loudness between the two different pitched tones, so we did not adjust the perceived loudness of them. However, we cannot rule out the possibility that this may have affected the results as we noted in the General Discussion. We have added the description about the importance of paying attention to the perceived loudness in the revised manuscript (L476∼478).

Comment #9:

In lines 179 – 181, the authors asked participants to imagine the designated visual features (i.e., brightness, vertical position, size, shape, or pattern) associated with each. Please clarify the purpose of the procedure.

Response:

First, crossmodal correspondences are supposed to be based on the relative rather than absolute differences between auditory and visual features (Brunetti et al., 2018; Spence, 2019). Therefore, the presentation of only one of a pair of tones during the rating phase is insufficient for participants to match the auditory feature with the visual feature. That’s why we included the previewing phase in the experiments.

Second, we asked participants to imagine the designated visual (or auditory) features associated with each of a pair of tones (or visual stimuli) presented during that phase. We believe that having an image driven by a visual/auditory stimulus can also work as a reference in a judgment of relative differences, which leads to higher credibility of the correspondence judgment. To make this clear, we have added this explanation in the General Discussion (L479∼487).

Comment #10:

It would be beneficial if authors could present the stimuli example of pairs of rounded–angular shapes, high SF and Low SF objects, and high and low positioned circles in Figure 1.

Response:

Following your suggestion, we have added Figure S1 to show the examples of the visual stimuli in the supporting information.

Comment #11:

In lines 206-210, the authors asked the participants to confirm whether they could distinguish the tones' auditory features. Please indicate the results of this experiment in belief.

Response: 

As we noted in response to your comment #4, the questions were used to confirm that they could distinguish the auditory features of the tones, and only those who answered both questions correctly were included in the analysis. We have added the exact number of participants who were unable to distinguish the features of the tones in the revised manuscript (L144∼146, L322∼324).

Comment #12:

Please indicate how much time it took from the start to the end of the experiment.

Response: 

Following your suggestion, we have added the information about the approximate time required to complete the experiment (L235∼236). 

Comment #13:

Results: In line 241, the authors wrote, "Two factors were extracted based on minimum average partial (MAP) criterion and a screen plot." Did the MAP and the screen plot extract the same number of factors? Where the two results differ, please state which result is preferred.

Response:

In both the experiments, the MAP scores consistently extracted two factors, and scree plots supported the extraction. We have modified the description in the revised manuscript (L268, L374∼75).

Comment #14:

Discussion: In lines 387 - 390, please make a rational explanation of why the distinction between structural and statistical correspondences has yet to be extracted in present studies.

Response:

Structural and statistical correspondences are supposed to be based on the neural connections and learning of natural regularities, respectively. However, while they may be different in the triggers that give rise to crossmodal correspondences, this does not mean that they have completely independent mechanisms. Indeed, it has been suggested that sensory experience may influence the development of neural structures. This may be the reason why the distinction between structural and statistical correspondences was not observed in the present study. To emphasize this point, we have modified the descriptions in the revised manuscript (L425∼429).

Comment #15:

In lines 411–413, the authors wrote, "…that the words "thin" and "thick" are more commonly used by people in Japan than the words "high" and "low" to describe stripe patterns that differ in spatial frequency." Please give references that support the statement.

Response:

In the Tsukuba Web Corpus (a corpus constructed from Japanese internet texts), the terms "thick" and "thin" frequently collocate with "stripe" as a modifier, whereas "high" and "low" do not appear. We have cited this reference as evidence to support the statement in the revised manuscript (L447).

Comment #16:

Lines 424–426, Please clarify how the participants’ interpretation has differed between the presentation order (auditory or visual stimuli first).

Response:

In the previewing phase of Experiment 1, participants were instructed to listen to auditory stimuli and to imagine the designated visual features associated with each stimulus. When the designated visual feature was size, they were instructed to imagine the size of visual objects associated with each stimulus. Since size is expressed "ookisa" in Japanese and the same word is used to describe the loudness of a tone, the participants might have interpreted the tones of different pitches as having different loudness levels by comparing them during the previewing stage. As a result, both the louder and high-pitched tones may have been judged to match larger stimuli in Experiment 1. In contrast, in the previewing phase of Experiment 2, participants were instructed to imagine the designated auditory features (i.e., loudness or pitch) associated with each of the visual stimuli presented. Therefore, participants could easily focus on the pitch when imagining the associated visual features. We have modified the description in the General Discussion to make this point clear (L470∼474).

---

## [Decision Letter · Decision Letter 1]

19 Oct 2023

PONE-D-23-11568R1How many categories are there in crossmodal correspondences?

A study based on exploratory factor analysisPLOS ONE

Dear Dr. Ohtake,

Thank you for submitting your manuscript to PLOS ONE. After careful consideration, we feel that it has merit but does not fully meet PLOS ONE’s publication criteria as it currently stands. Therefore, we invite you to submit a revised version of the manuscript that addresses the points raised during the review process.

We look forward to receiving your revised manuscript.

Kind regards,

Kyoshiro Sasaki, Ph.D.

Academic Editor

PLOS ONE

Journal Requirements:

Additional Editor Comments:

Thank you for revising the manuscript. Both reviewers were satisfied with your revisions, although Reviewer 1 pointed out only minor issues. Therefore, my decision is to request a minor revision. Please check the points raised by the reviewer.

Reviewers' comments:

Reviewer's Responses to Questions

**Comments to the Author**

1. If the authors have adequately addressed your comments raised in a previous round of review and you feel that this manuscript is now acceptable for publication, you may indicate that here to bypass the “Comments to the Author” section, enter your conflict of interest statement in the “Confidential to Editor” section, and submit your "Accept" recommendation.

Reviewer #1: (No Response)

Reviewer #3: All comments have been addressed

2. Is the manuscript technically sound, and do the data support the conclusions?

Reviewer #1: Yes

Reviewer #3: Yes

3. Has the statistical analysis been performed appropriately and rigorously? 

Reviewer #1: I Don't Know

Reviewer #3: Yes

4. Have the authors made all data underlying the findings in their manuscript fully available?

Reviewer #1: Yes

Reviewer #3: Yes

5. Is the manuscript presented in an intelligible fashion and written in standard English?

Reviewer #1: Yes

Reviewer #3: Yes

6. Review Comments to the Author

Reviewer #1: MS improved in revision:

Final minor corrections/improvements =

Mudd (1963) didn’t investigate CCs as such. Rather they investigated spatial qualities associated with tones, that subsequent researchers have interpreted in terms of CCs.

Smith is not an author of Marks et al.’s (1987) paper, but write a separate commentary on their article.

There are still problems with references. Newly-Added ref 1, you need to make clear these are (Eds.) of handbook not authors of it.

Sadaghiani, S., Maier, J. X., & Noppeney, U. (2009). Natural, metaphoric, and linguistic auditory direction signals have distinct influences on visual motion processing. Journal of Neuroscience, 29, 6490-6499. May also be relevant in showing different neural substrates for linguistic and non-linguistic correspondences

p. 21 “For example, the

102 correspondence between pitch and elevation could be explained either by the

103 internalization of natural statistics or the use of the same words” I am not sure it is necessarily either/or, various correspondences might also have some contribution from both?

Reviewer #3: I have thoroughly examined the revised paper.

The revision has clarified the primary message of this manuscript and reads much better. Most of the questions made by the reviewer were answered satisfactory.

I recommend the paper be published in its present form.

7. PLOS authors have the option to publish the peer review history of their article (what does this mean?). If published, this will include your full peer review and any attached files.

Reviewer #1: No

Reviewer #3: No

---

## [Author Response · Author response to Decision Letter 1]

25 Oct 2023

Response to Reviewer #1

We greatly appreciate your invaluable support for our manuscript. Here, we summarize how we addressed your suggestions and revised the manuscript.

Comment #1

Mudd (1963) didn’t investigate CCs as such. Rather they investigated spatial qualities associated with tones, that subsequent researchers have interpreted in terms of CCs.

Response:

As you pointed out, Mudd (1963) investigated spatial features associated with pure tones in four dimensions. To accurately introduce the study, we have revised the description (L51 ∼ 56). 

Comment #2:

Smith is not an author of Marks et al.’s (1987) paper, but write a separate commentary on their article.

Response:

Thank you for pointing this out. We have corrected the citation of the relevant text and References (L60 ∼ 61, ref 8). 

Comment #3:

There are still problems with references. Newly-Added ref 1, you need to make clear these are (Eds.) of handbook not authors of it.

Response:

Thank you for your attention to detail. We have corrected the mistake in the Reference according to the PLOS reference style (ref 1).

Comment #4:

Sadaghiani, S., Maier, J. X., & Noppeney, U. (2009). Natural, metaphoric, and linguistic auditory direction signals have distinct influences on visual motion processing. Journal of Neuroscience, 29, 6490-6499. May also be relevant in showing different neural substrates for linguistic and non-linguistic correspondences

Response:

Thank you for sharing the important paper. We have cited this paper in the relevant text in the General Discussion (L515).

Comment #5:

p. 21 “For example, the correspondence between pitch and elevation could be explained either by the internalization of natural statistics or the use of the same words” I am not sure it is necessarily either/or, various correspondences might also have some contribution from both?

Response:

We meant to say that pitch and elevation could be explained at least in two ways. As you pointed out, we did not mean that the two explanations/underlying processes were exclusive. To make this clear, we have modified the description (L103).

---

## [Editor Report · Decision Letter 2]

26 Oct 2023

How many categories are there in crossmodal correspondences?

A study based on exploratory factor analysis

PONE-D-23-11568R2

Dear Dr. Ohtake,

We’re pleased to inform you that your manuscript has been judged scientifically suitable for publication and will be formally accepted for publication once it meets all outstanding technical requirements.

Kind regards,

Kyoshiro Sasaki, Ph.D.

Academic Editor

PLOS ONE

---

## [Editor Report · Acceptance letter]

6 Nov 2023

PONE-D-23-11568R2 

How many categories are there in crossmodal correspondences?
A study based on exploratory factor analysis 

Dear Dr. Ohtake:

I'm pleased to inform you that your manuscript has been deemed suitable for publication in PLOS ONE. Congratulations! Your manuscript is now with our production department. 

Kind regards, 

on behalf of

Dr. Kyoshiro Sasaki 

Academic Editor

PLOS ONE